# Unveiling the Multifaceted Dynamics of Breast Cancer: A Copula Regression Approach to Modeling and Predicting Outcomes

Huma Rani[1][*], Tahir Mehmood[2], Muhammad Aslam[1¤], Laila A. Al-Essa[3]

1 Department of Mathematics and Statistics, Riphah International University, Islamabad, Pakistan,
2 School of Natural Sciences (SNS), National University of Sciences and Technology (NUST), Islamabad, Pakistan, 3 Department of Mathematical Sciences, College of Science, Princess Nourah bint Abdulrahman University, Riyadh, Saudi Arabia

⊙ These authors contributed equally to this work.
¤ Current address: Department of Statistics, Riphah International University, Islamabad, Pakistan
* huma_waleed2014@hotmail.com

## Abstract

This study examines the application of flexible copula regression models to analyze the complex interdependencies among clinical variables in breast cancer data. As the most commonly diagnosed cancer and the second leading cause of cancer-related deaths among women worldwide, breast cancer presents both clinical and analytical challenges. Unlike traditional multivariate approaches, copulas offer greater flexibility in capturing complex, nonlinear, and asymmetric dependencies between mixed-type outcomes. The present study examines copula-based regression models to investigate the joint behavior of clinical variables in patients with breast cancer. We explored multiple copula families to jointly model overall survival (binary) and age at diagnosis (continuous) in the METABRIC dataset. Goodness-of-fit metrics guide model comparison and selection, with the Gumbel copula demonstrating superior performance in capturing the upper tail dependence associated with favorable outcomes, such as younger age and improved survival. Formal model comparison against an independent margins baseline confirmed that accounting for dependence via a copula significantly improves model fit (likelihood ratio test: $\chi^2 = 2190.24$, df $= 1$, $p < 0.0001$), and PIT diagnostics validated the adequacy of both marginal specifications. The findings support the integration of copula models into clinical research, facilitating a more nuanced understanding of cancer progression and enabling more accurate risk assessment and data-driven decision-making in oncology.

## Introduction

Copula regression has emerged as a powerful tool for modelling dependence structures in multivariate data, providing a flexible framework for constructing joint distributions when outcomes follow non-normal or mixed distributions [1,2]. Traditional correlation

**Data availability statement:** The data that support the findings of this study is available in Figshare at https://figshare.com/s/1347711e0e6d2a2bc6fe, and will be publicly available after the acceptance of this article.

**Funding:** Princess Nourah bint Abdulrahman University Researchers Supporting Project number (PNURSP2026R443), Princess Nourah bint Abdulrahman University, Riyadh, Saudi Arabia.

**Competing interests:** The authors have declared that no competing interests exist.

measures often fail to capture complex dependencies, particularly in non-Gaussian settings and tail dependence [3], making copula theory essential for accurate multivariate analysis. The most attractive feature of copula modeling is that parameter estimation and inference can be performed using standard likelihood procedures [4].

Copula methods have seen widespread adoption across diverse scientific fields due to their flexibility in capturing complex dependence structures. In finance and economics, they model asymmetric exchange rate dependencies [5] and income-consumption relationships [6]. Environmental scientists employ copulas for analyzing concurrent extreme weather events [7,8] and hydrological droughts [9]. Transportation researchers use them to jointly analyze incident clearance and response times [10], while agricultural economists model crop yield dependencies [11]. Different copula models have been proposed, utilizing innovative approaches, and applied to various datasets. [12] applied copulas in health economics, while [13] developed copula methods for correlated survival data. In genomics, [14] used copulas to model gene dependencies, and [15] applied them to construct biological networks. Recent biomedical applications include modeling breast cancer metastasis patterns [16], survival prediction under dependent censoring [17], and joint analysis of mixed discrete-continuous outcomes [18]. Copula models are also discussed by [19,20], but their application to clinical biostatistics, especially in cancer research, remains underdeveloped.

This gap is particularly salient in breast cancer studies, where key outcomes and predictors may exhibit complex dependence patterns that traditional regression approaches (e.g., logistic or generalized linear regression) fail to capture. Breast cancer remains the most commonly diagnosed malignancy and the second leading cause of cancer-related mortality in women worldwide, with approximately 2.3 million new cases and 685,000 deaths annually [21]. Age at diagnosis is a critical prognostic factor [22], with survival patterns showing significant variation across age groups and geographical regions [23,24]. These classical regression models assume independence between outcomes and often obscure clinically important dependence structures and relationships. This article explores the copula approach to model the breast cancer dataset. Despite advances in screening and treatment for primary prevention, early diagnosis, and survival prediction have improved outcomes, but further methodological improvements in capturing complex dependencies are still needed.

In the present study, we address the methodological challenge in two ways. First, by properly handling mixed discrete and continuous outcomes, and second, by incorporating interpretable dependence measures through copula models that are required for clinical decision-making. Previous approaches relied on restrictive distributional assumptions [25] or on either modeling outcomes separately (losing dependence information). Our approach builds on recent advances in copula regression for mixed outcomes [18,26] while addressing the specific needs of cancer research.

Our work makes two key statistical contributions:

- We demonstrate how copula regression models with latent variable probit margins can effectively model the dependence between binary and continuous outcomes in breast cancer data, thereby overcoming the limitations of previous methods.

- We provide a reproducible framework for testing and comparing alternative dependence structures through information criteria in cancer research.

Using data from the METABRIC (Molecular Taxonomy of Breast Cancer International Consortium) cohort (n = 1,904), we show how copula methods reveal clinically meaningful dependence between the chosen pair of response variables (binary-continuous), i.e., overall survival($y_1$), and age at diagnosis($y_2$), accounting for the influence of covariates, using various copula families. Importantly, we model age at diagnosis and survival as joint manifestations of the disease process rather than causally linked variables, allowing estimation of their residual association after covariate adjustment, an approach particularly valuable for identifying patient subgroups with unusual age-survival patterns. The implementation is through the R package GJRM (Generalized Joint Regression Modeling) [27].

We develop a template for cancer research by characterizing the interplay between treatment response and age stratification.

The rest of the paper is organized as follows. Section II describes the data and study variables included in the analysis. Section III presents the mathematical framework of the bivariate copula models with mixed binary–continuous margins and associated likelihood formulations. Section IV presents the empirical results, along with a discussion and conclusions. Additional materials and technical derivations are provided in the Supporting Information Appendix section.

## Materials and methods

### Data source

The *Molecular Taxonomy of Breast Cancer International Consortium (METABRIC)* is a large-scale, retrospective study jointly conducted by research teams in Canada and the United Kingdom. It provides comprehensive genomic and clinical profiles for over 2,000 breast cancer patients [28]. The dataset used in this study was obtained from *Kaggle*, with the original source being *cBioPortal for Cancer Genomics*. For the present analysis, only clinical and demographic variables were retained, while genomic features were excluded to focus on clinical predictors of breast cancer outcomes. Data preprocessing involved imputing missing values using the mean for continuous variables and the mode for categorical variables. Redundant or irrelevant features were removed prior to analysis, resulting in a final dataset comprising 1,904 patients and 30 clinical variables.

### Study variables

The analysis focused on two primary response variables: 1) a binary overall survival indicator, and 2) a continuous age at diagnosis. Clinical covariates were selected based on their known prognostic relevance in breast cancer (Fig 1).

- **Overall Survival ($y_1$)**: A binary survival indicator as provided in the METABRIC dataset. The variable is coded as 0 if the patient was deceased and 1 if the patient was alive at the last clinical follow-up, irrespective of cause of death. This coding was verified by cross-referencing with the descriptive variable `death_from_cancer`. The binary formulation was adopted to facilitate the application of the copula framework to mixed binary–continuous outcomes, which is the primary methodological focus of this study.

- **Age at Diagnosis ($y_2$)**: A continuous variable (in years) representing the patient's age at the time of breast cancer diagnosis. Age is a well-established prognostic factor in breast cancer, although its relationship with disease aggressiveness and survival is complex and population-dependent [22,23]. In this study, age is treated as a marginal response solely to enable joint modeling with survival status and to quantify their dependence structure within the METABRIC cohort. No causal or temporal interpretation is implied by this specification.

- **Clinical Covariates**: The model adjusted for the following prognostic factors:

  - **Type of breast surgery**: breast-conserving vs mastectomy.

  - **Cellularity**: low, moderate, high.

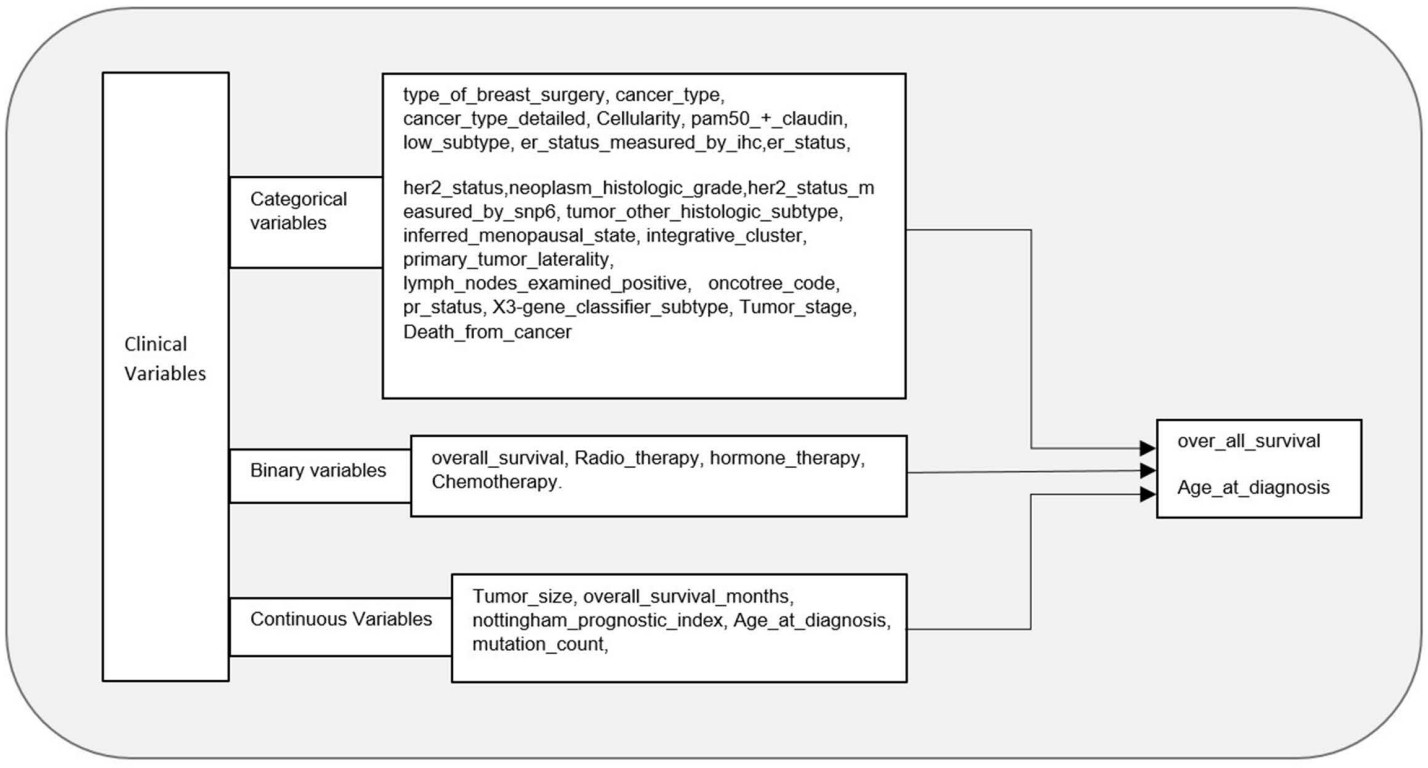

**Fig 1. Potential Risk Factors for Breast Cancer: Clinical Variables in the METABRIC Cohort.**

- **ER, PR, and HER2 status**: positive vs negative.

- **Neoplasm histologic grade (NHG)**: ordinal (grades 1–4, plus x).

- **Other histologic subtype (OHS)**: ductal/NST, lobular, mixed, mucinous, medullary, metaplastic, other.

- **Tumor laterality**: left vs right.

- **Integrative cluster (IC)**: subgroups 1–10.

- **Treatment variables**: chemotherapy, radiotherapy, and hormone therapy (all binary).

- **Mutation count**: numeric.

- **Nottingham Prognostic Index (NPI)**: numeric index combining tumor size, grade, and lymph node status.

- **Menopausal state**: pre vs post.

- **Tumor size**: continuous, measured in mm.

## Methodology

### Copula-based modeling framework

The joint modeling framework in this study is based on Sklar's theorem [29], which states that any multivariate distribution can be decomposed into its marginal distributions and a copula function that captures the dependence structure. This makes copula models particularly suitable for analyzing outcomes with different distributional forms.

We focus on a bivariate setting where the two outcomes of interest are: (i) a binary response representing overall survival ($Y_1$), and (ii) a continuous response representing age at diagnosis ($Y_2$). The binary outcome $Y_1$ represents overall survival status and is defined as

$$Y_1 = \begin{cases} 1, & \text{if the patient was alive at the last available clinical follow-up,} \\ 0, & \text{if the patient was deceased, irrespective of cause of death.} \end{cases}$$

Although time-to-event information is available in the METABRIC cohort, a binary formulation is adopted here to align with the primary methodological objective of this study, namely, copula-based joint modeling of mixed binary–continuous outcomes rather than hazard-based survival analysis.

Age at diagnosis ($Y_2$) is included as a continuous marginal outcome solely to enable joint modeling and to quantify its dependence with survival status. It is not interpreted as being predicted by survival, nor does its inclusion imply any temporal or causal direction between the two outcomes.

The joint distribution of these outcomes can be expressed as

$$F(y_1, y_2) = C(F_1(y_1), F_2(y_2); \theta),$$

(1)

where $F_1$ and $F_2$ are the marginal cumulative distribution functions (CDFs) of $Y_1$ and $Y_2$, respectively, $C(\cdot)$ is a copula function, and $\theta$ is the parameter capturing the strength and shape of dependence.

**Modeling mixed binary and continuous outcome.** Directly applying copulas to a mixed binary-continuous case is not straightforward because the copula function is not unique when a marginal distribution is discrete. To address this, we adopted a latent variable approach following [30]. We assume that the binary survival outcome $Y_1$ is determined by an underlying, continuous latent variable $Y_1^*$ such that:

$$Y_1 = \mathbb{I}(Y_1^* > 0),$$

where $\mathbb{I}(\cdot)$ is the indicator function. We assume $Y_1^*$ follows a standard normal distribution, which inherently leads to a probit link for the probability of survival.

This formulation allows us to derive the joint likelihood for an observation $(y_1, y_2)$. The resulting joint probability mass and density function is:

$$f(y_1, y_2) = \left[ \frac{\partial C(F_1^*(0), F_2(y_2))}{\partial F_2(y_2)} \right]^{(1-y_1)} \left[ 1 - \frac{\partial C(F_1^*(0), F_2(y_2))}{\partial F_2(y_2)} \right]^{y_1} \cdot f_2(y_2),$$

(2)

where $f_2(y_2)$ is the marginal density of $Y_2$ (age at diagnosis), and the partial derivative represents the conditional distribution of the latent variable $Y_1^*$ given the observed age $Y_2$.

Here, Equation (2) defines the joint probability structure implied by the copula and characterizes a symmetric dependence between the latent survival process and age at diagnosis. It is critical to note that this formulation, and the copula model in general, is symmetric with respect to $Y_1$ and $Y_2$. The model estimates the stochastic dependence between the two outcomes after accounting for covariates, and the copula parameter $\theta$ quantifies this association without implying that survival "explains" or "predicts" age, or vice versa. The conditional representation appearing in the likelihood is a mathematical device required for estimation in mixed discrete–continuous settings and does not denote any causal, temporal, or directional relationship.

**Specification of marginal distributions.**

• For the **binary survival outcome** $Y_1$, the latent variable approach with a probit link was used, as described above.

- For the **continuous age outcome** $Y_2$, several candidate distributions were evaluated, including the normal, gamma, log-normal, logistic, and inverse gamma etc. Model comparison based on Akaike and Bayesian Information Criteria (AIC/BIC) indicated that the normal distribution provided the best fit for our data (see S1 Table and S1 Fig in Supporting Information).

**Copula functions and dependence measurement.** To capture the dependence between survival and age, we considered both elliptical and Archimedean copula families. Specifically, we examined the Gaussian, Clayton, Gumbel, and Frank copulas, which represent distinct forms of dependence, including symmetric, lower-tail, upper-tail, and central association [31,32]. The copula parameter $\theta$ was transformed into Kendall's $\tau$, a rank-based correlation coefficient ranging from −1 to 1. This provides an intuitive, standardized measure of the association strength between survival and age, facilitating interpretation and comparison across copulas.

**Regression framework and estimation.** We embedded the joint model within a regression framework to relate covariates to all parameters of the distribution. Specifically, we defined additive predictors for each parameter:

- The probability of survival ($\pi$) was linked to covariates via a probit function.
- The mean ($\mu$) of the age distribution was modeled with a linear predictor.
- The variance ($\sigma^2$) of the age distribution was modeled with a log-link to ensure positivity.
- The copula dependence parameter ($\theta$) was modeled as a constant parameter within each copula family, capturing the overall strength of dependence between the two outcomes. Although the general modeling framework allows the copula dependence parameter $\theta$ to link to covariates, all results reported in this study correspond to models with a constant $\theta$ within each copula family. Covariate-dependent dependence structures were explored only in preliminary analyses and are beyond the scope of the present paper.

More formally, let $\vartheta_i = (\pi_i, \mu_i, \sigma_i, \theta)^\top$ denote the parameter vector for patient $i$. The model is specified through the following regression structures:

**For the binary survival margin ($\pi$):**

$$\Phi^{-1}(\pi_i) = \beta_0^\pi + \beta_1^\pi \cdot \text{Age at Diagnosis}_i + \beta_2^\pi \cdot \text{ER Status}_i + \beta_3^\pi \cdot \text{PR Status}_i + \beta_4^\pi \cdot \text{HER2 Status}_i$$

$$+\beta_5^\pi \cdot \text{Chemotherapy}_i + \beta_6^\pi \cdot \text{Hormone Therapy}_i + \beta_7^\pi \cdot \text{Radiotherapy}_i+$$

$$\beta_8^\pi \cdot \text{NPI}_i + \beta_9^\pi \cdot \text{Positive Lymph Nodes}_i + \beta_{10}^\pi \cdot \text{Tumor Size}_i$$

**For the continuous age mean ($\mu$):**

$$\mu_i = \beta_0^\mu + \beta_1^\mu \cdot \text{Overall Survival}_i + \beta_2^\mu \cdot \text{Histologic Grade2}_i + \beta_3^\mu \cdot \text{Histologic Grade3}_i + \beta_4^\mu \cdot \text{NPI}_i$$

$$+\beta_5^\mu \cdot \text{Tumor Size}_i + \beta_6^\mu \cdot \text{Positive Lymph Nodes}_i + \beta_7^\mu \cdot \text{ER status}_i + \beta_8^\mu \cdot \text{PR Status}_i$$

**For the age variance ($\sigma^2$):**

$$\log(\sigma_i) = \gamma_0$$

where the log link ensures positivity, and only an intercept is included.

**For the copula dependence ($\theta$):**

$$\theta = \theta_0 \quad \text{(estimated separately for each copula family)}$$

This formulation explicitly specifies which covariates enter each component of the model and confirms that the copula dependence parameter $\theta$ is constant (i.e., no covariates) across all reported models.

The model parameters were estimated simultaneously by maximizing the log-likelihood function, which for a sample of $n$ independent observations is:

$$\ell(\beta) = \sum_{i=1}^{n} \left[ (1 - y_{i1}) \log\{F_{1|2}(0|y_{i2})\} + y_{i1} \log\{1 - F_{1|2}(0|y_{i2})\} + \log\{f_2(y_{i2})\} \right],$$

(3)

where $F_{1|2}(0|y_{i2})$ is the conditional probability from Equation (2). Parameter estimation was performed using numerical optimization. Full details of the likelihood derivatives and the regression formulation are provided in S1 Appendix.

### Ethics statement

This study used publicly available, de-identified data from the *Molecular Taxonomy of Breast Cancer International Consortium (METABRIC)* project [28]. The dataset was accessed on August 24, 2023, via *Kaggle* (originally from the *cBioPortal for Cancer Genomics*). As this analysis utilized previously collected and fully anonymized data, no direct patient contact or intervention occurred, and no identifiable personal information was accessed. Therefore, institutional review board (IRB) approval and informed consent were not required. The original METABRIC study received ethics approval from relevant institutional review boards in the United Kingdom and Canada, as reported in [28].

## Results and discussion

### Descriptive analysis of clinical covariates

To provide an overview of the study population, we first conducted a descriptive analysis of the METABRIC cohort to characterize the demographic, clinical, and molecular features of patients with breast cancer. This summary lays the groundwork for understanding the variability within the dataset and supports the subsequent modeling and copula-based analyses. Table 1 summarizes the clinical and molecular characteristics of the METABRIC breast cancer cohort ($N = 1{,}904$). Patients were diagnosed at a mean age of 61 years (±13), with tumors typically measuring around 26 mm in size. On average, two lymph nodes were positive, though the distribution was highly skewed, reflecting a wide range of disease progression.

Most tumors were stage 2 (68.3%) and classified as high-grade (grade 3 in 52.5% of cases), indicating a tendency toward aggressive disease biology in this cohort. Cellular morphology was most frequently reported as high (52.2%), further supporting this pattern.

In terms of molecular markers, ER-positive status was the most common (76.6%), followed by PR-positive (53.0%) and HER2-positive (12.4%) tumors. These distributions are consistent with typical hormone-receptor-driven breast cancer. Treatment data showed that nearly two-thirds of patients received hormone therapy (61.7%), and a similar proportion underwent radiotherapy (59.7%). Chemotherapy was administered to approximately one-fifth of patients (20.8%).

Finally, integrative molecular clustering revealed substantial heterogeneity. IntClust 8 (15.2%) and IntClust 3 (14.8%) were the most frequent subtypes, reflecting the complex genomic architecture of breast cancer. Further, various key factors of breast cancer data are visualized to examine their impact on survival status and age by groups. Only a few key covariates are chosen for descriptive analysis through bar graphs. Fig 2 displays both response variables, i.e., age at diagnosis (groups) and overall survival, in a single graph. It is evident from the graph that the risk of not surviving the disease increases with age. The proportion of deceased patients increased markedly in the 60–79 and 80–100 year groups, reflecting the impact of age on breast cancer prognosis.

The comparative analysis of clinical and molecular features by survival status is illustrated in Figs 3–6.

**Table 1. Descriptive statistics of the METABRIC cohort (N = 1,904). Continuous variables are reported as mean ± SD; categorical variables as n (%).**

| Variable | Level | n | % / Mean ± SD |
|---|---|---|---|
| **Continuous variables** | | | |
| Age at Diagnosis (years) | – | – | 61.1 ± 13.0 |
| Tumor Size (mm) | – | – | 26.2 ± 15.1 |
| Positive Lymph Nodes | – | – | 2.0 ± 4.1 |
| Mutation Count | – | – | 5.7 ± 4.0 |
| Nottingham Prognostic Index (NPI) | – | – | 4.0 ± 1.1 |
| **Categorical variables** | | | |
| Overall Survival | Alive (1) | 801 | 42.1 |
| | Deceased (0) | 1103 | 57.9 |
| Tumor Stage | 0 | 4 | 0.2 |
| | 1 | 475 | 24.9 |
| | 2 | 1301 | 68.3 |
| | 3 | 115 | 6.0 |
| | 4 | 9 | 0.5 |
| Histologic Grade | 1 | 165 | 8.7 |
| | 2 | 740 | 38.9 |
| | 3 | 999 | 52.5 |
| Cellularity | High | 993 | 52.2 |
| | Moderate | 711 | 37.3 |
| | Low | 200 | 10.5 |
| ER Status | Positive | 1459 | 76.6 |
| | Negative | 445 | 23.4 |
| PR Status | Positive | 1009 | 53.0 |
| | Negative | 895 | 47.0 |
| HER2 Status | Negative | 1668 | 87.6 |
| | Positive | 236 | 12.4 |
| Chemotherapy | Yes | 396 | 20.8 |
| | No | 1508 | 79.2 |
| Radiotherapy | Yes | 1137 | 59.7 |
| | No | 767 | 40.3 |
| Hormone Therapy | Yes | 1174 | 61.7 |
| | No | 730 | 38.3 |
| Integrative Cluster | IC1 | 132 | 6.9 |
| | IC2 | 72 | 3.8 |
| | IC3 | 282 | 14.8 |
| | IC4ER- | 74 | 3.9 |
| | IC4ER+ | 244 | 12.8 |
| | IC5 | 184 | 9.7 |
| | IC6 | 84 | 4.4 |
| | IC7 | 182 | 9.6 |
| | IC8 | 289 | 15.2 |
| | IC9 | 142 | 7.5 |
| | IC10 | 219 | 11.5 |

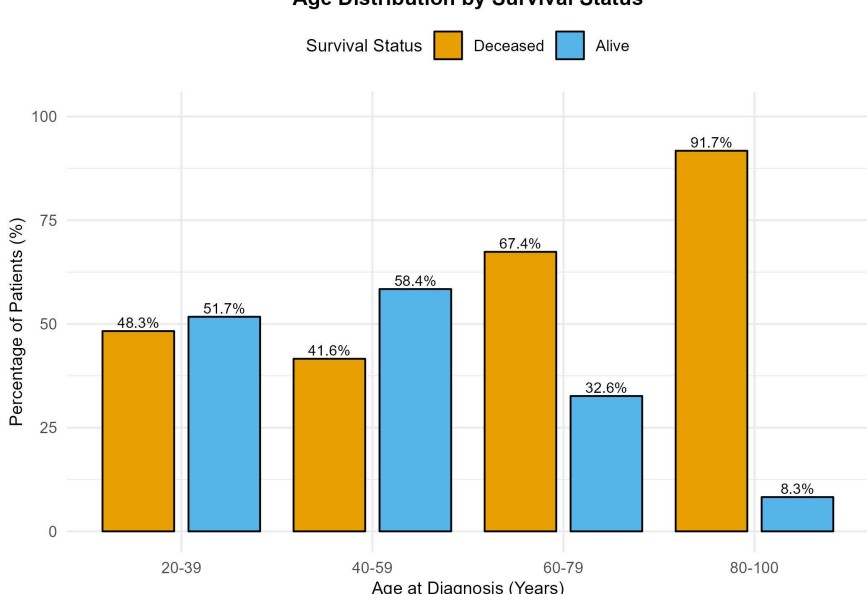

**Fig 2. Survival status across age groups in the METABRIC cohort.** Older patients show higher mortality, while survival is more common among younger groups, underscoring the strong association between advanced age and poorer survival outcomes.

The Fig 3 illustrates the distribution of major treatment types, i.e., chemotherapy, hormone therapy, and radiotherapy, by survival status. A larger proportion of patients who received chemotherapy or hormone therapy did not survive, while many survivors did not undergo radiotherapy. These patterns suggest that treatment response and survival outcomes vary across therapies, but they do not imply a direct causal relationship.

Fig 4 illustrates the cellularity, comprising its three key factors. The highest bars, indicating the "high" factor, signify that patients did not survive the disease diagnosed with high and moderate cellularity. Still, these factors also demonstrate the second-highest survival rate among patients with the disease.

According to the findings in Fig 5, the factor of tumor histologic subtype that grasps the utmost significance is Ductal/NST. This factor was diagnosed in patients who did not survive the cancer as well as those who did.

Fig 6 shows that patients in integrative clusters "3" and "8" had lower survival rates, while those in cluster "*4ER⁺*" showed noticeably better survival outcomes.

Additional visualizations are provided in the Supporting Information (S2–S5 Figs) to complement the main descriptive findings. S2 Fig explores surgical type and tumor laterality by survival. S3 Fig highlights that higher tumor cellularity tends to occur among older patients, whereas S4 Fig illustrates age-related variation in integrative molecular clusters. Finally, S5 Fig presents histologic subtype distribution by age, confirming that ductal/NST carcinoma is the predominant histologic subtype across all groups.

## Correlation structure of clinical covariates

The dependency structure among the METABRIC clinical variables was characterized using a mixed-type correlation matrix (Fig 7). Since the dataset included continuous, binary, and categorical variables, appropriate statistical measures were applied for each pair: Pearson's correlation for continuous pairs, Cramér's V for categorical pairs, and point-biserial correlations for continuous-binary pairs. For multi-level categorical variables paired with continuous measures,

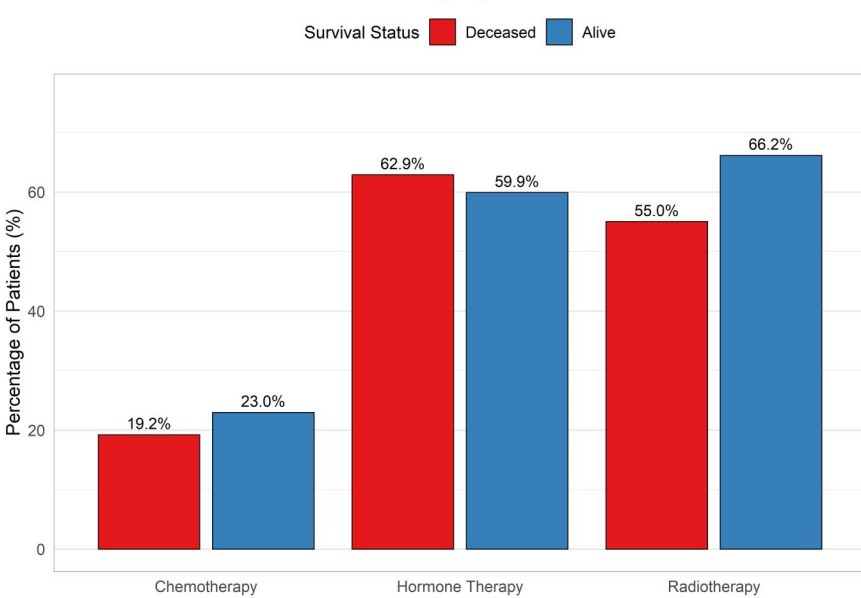

**Fig 3. Distribution of treatment types stratified by survival status.** Chemotherapy, radiotherapy, and hormone therapy exhibit distinct patterns across deceased and surviving patients.

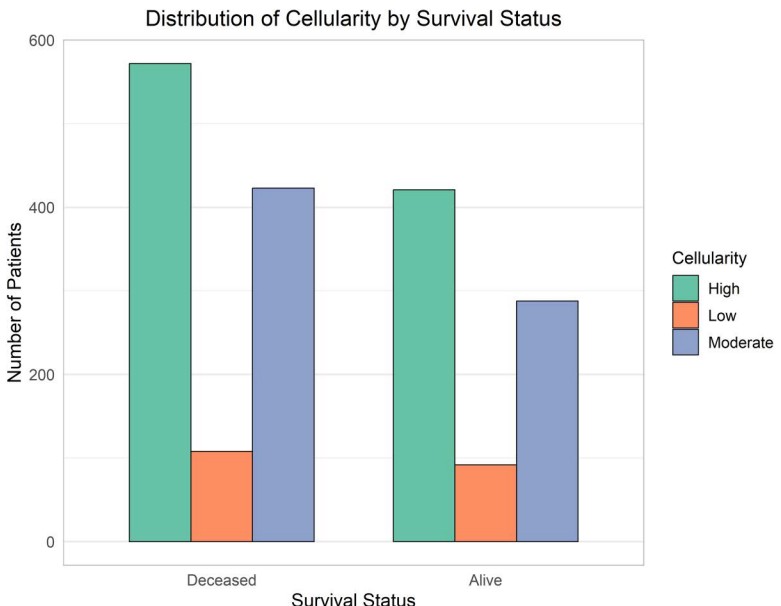

**Fig 4. Association between tumor cellularity and survival status.** Higher cellularity appears more common among non-survivors, suggesting aggressive tumor phenotypes.

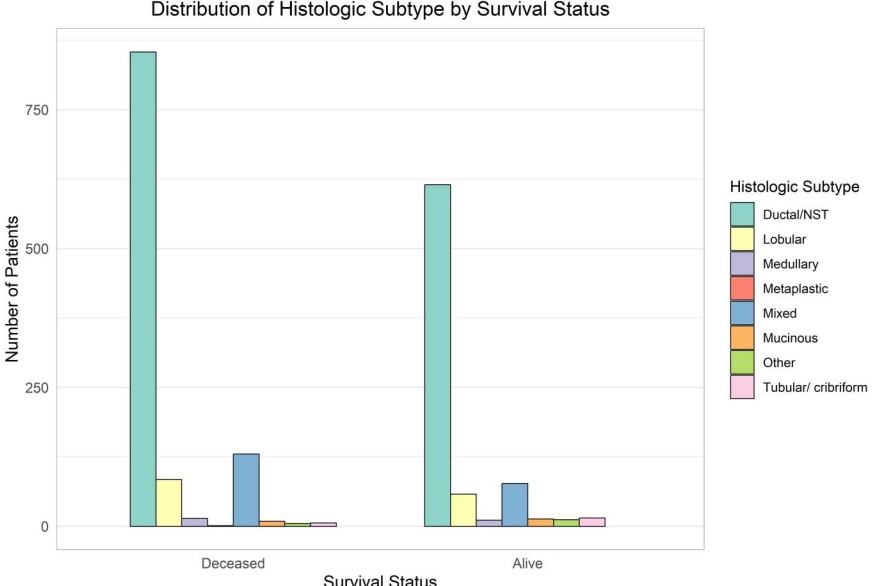

**Fig 5. Distribution of histologic subtypes among survival groups.** Invasive ductal/NST carcinoma dominates both groups.

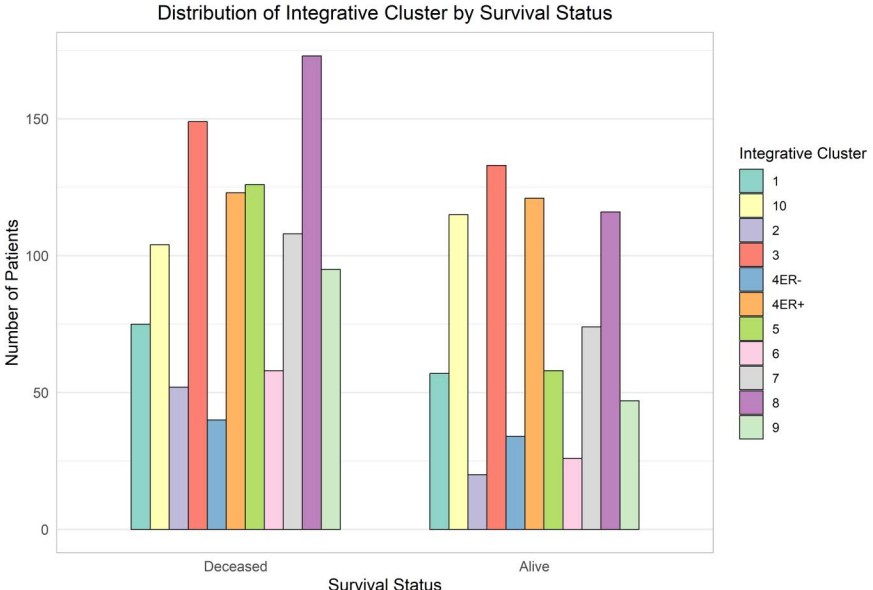

**Fig 6. Variation in integrative molecular clusters by survival status.** Specific clusters show a higher mortality association, reflecting molecular heterogeneity.

ANOVA-based $\eta^2$ approximations were used. The resulting symmetric matrix allows direct visual comparisons across variable types.

As shown in Fig 7, several patterns emerged. Lymph node status and the Nottingham Prognostic Index (NPI) were strongly associated ($r \approx 0.56$), as were tumor grade and NPI ($r \approx 0.59$), which is expected since both contribute to NPI

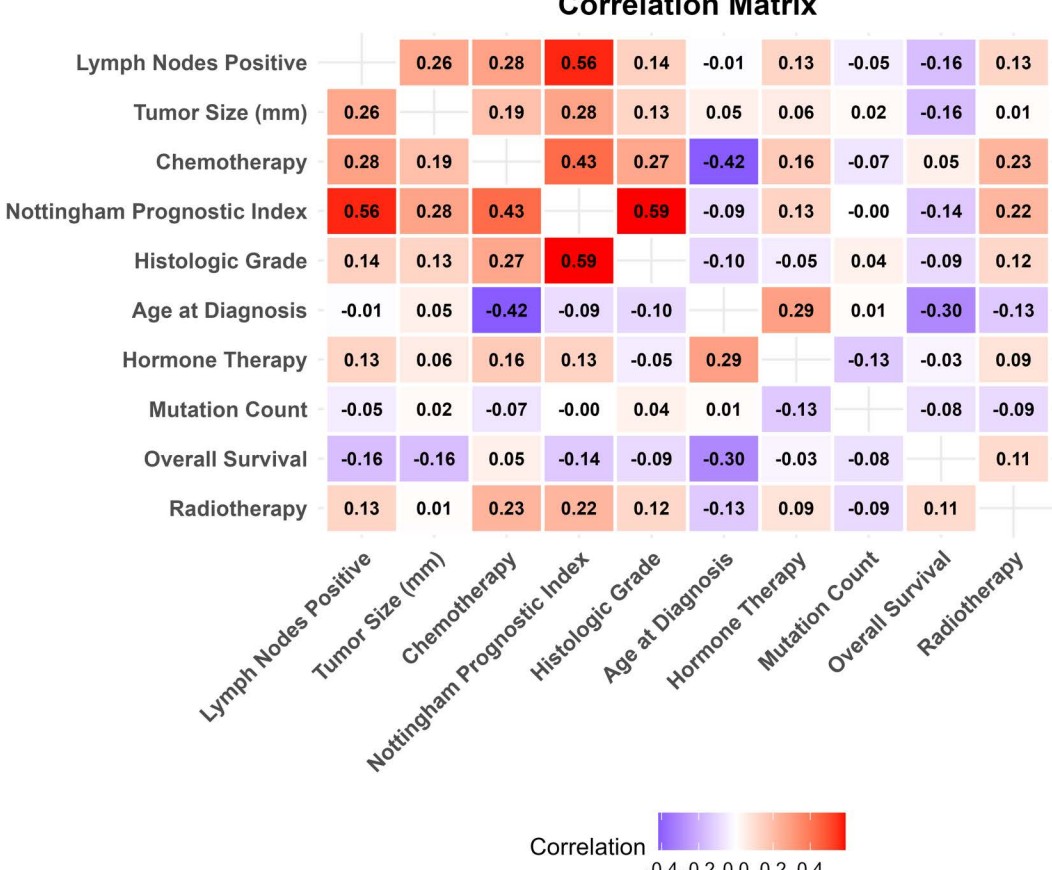

**Fig 7. Mixed-type correlation heatmap of key clinical variables in the METABRIC breast cancer dataset.** Correlations are computed using Pearson's *r*, Cramér's *V*, and point-biserial or ANOVA-based methods, depending on variable type. Strong associations are shown in red, while weaker or negative associations are displayed in purple or white.

calculation. Tumor size showed moderate correlation with both NPI ($r \approx 0.28$) and lymph node status, reinforcing its role in disease progression.

In contrast, treatment-related variables (chemotherapy, hormone therapy, radiotherapy) exhibited weak correlations with prognostic indicators. Age at diagnosis showed minimal associations, with a weak inverse correlation with NPI ($r \approx -0.30$) and with overall survival ($r \approx -0.14$), suggesting that age alone is not a strong driver of survival differences in this cohort. This correlation analysis supports the inclusion of both prognostic and biological factors in the joint model. It also demonstrates how a mixed correlation matrix helps identify overlapping variables and guide the selection of key predictors for copula-based regression. As part of this exploratory analysis, several candidate outcome pairings were examined. Joint associations between overall survival and tumor size, as well as between overall survival and the Nottingham Prognostic Index, were weak and close to zero. In contrast, the association between age at diagnosis and overall survival, while modest, was consistently non-negligible. This observation motivated the selection of the age–survival pairing for subsequent copula-based joint modeling.

## Interpretation of copula model results

To jointly model the binary response variable overall survival ($Y_1$) and the continuous response variable age at diagnosis ($Y_2$), we fitted copula-based regression models using Gaussian, Gumbel, Frank, and Clayton copulas. The models account for dependencies between the two outcomes while allowing distinct marginal specifications, a probit model for $Y_1$ and an identity link for $Y_2$. Further, the Joint densities and contour plots of fitted copulas are shown in Fig 8. As outlined in the Methods, this symmetric formulation is intended to quantify statistical association rather than causal relationships between the outcomes. Accordingly, age at diagnosis is treated as a joint outcome to characterize its empirical dependence with survival, not as a temporally causal response.

Table 2 summarizes the results of the bivariate regression model with Gaussian and Gumbel copula functions for clinical variables of the METABRIC data. The Gaussian copula model revealed that age at diagnosis was negatively associated with survival probability (Estimate = −0.085, $p < 0.001$), indicating that younger patients were more likely to survive. Similarly, ER-positive status was associated with a significant increase in survival (Estimate = 0.765, $p < 0.001$). However, PR status, HER2 status, and treatment-related covariates were not significant predictors. In the model for age, overall survival showed a strong inverse association (Estimate = −7.898,

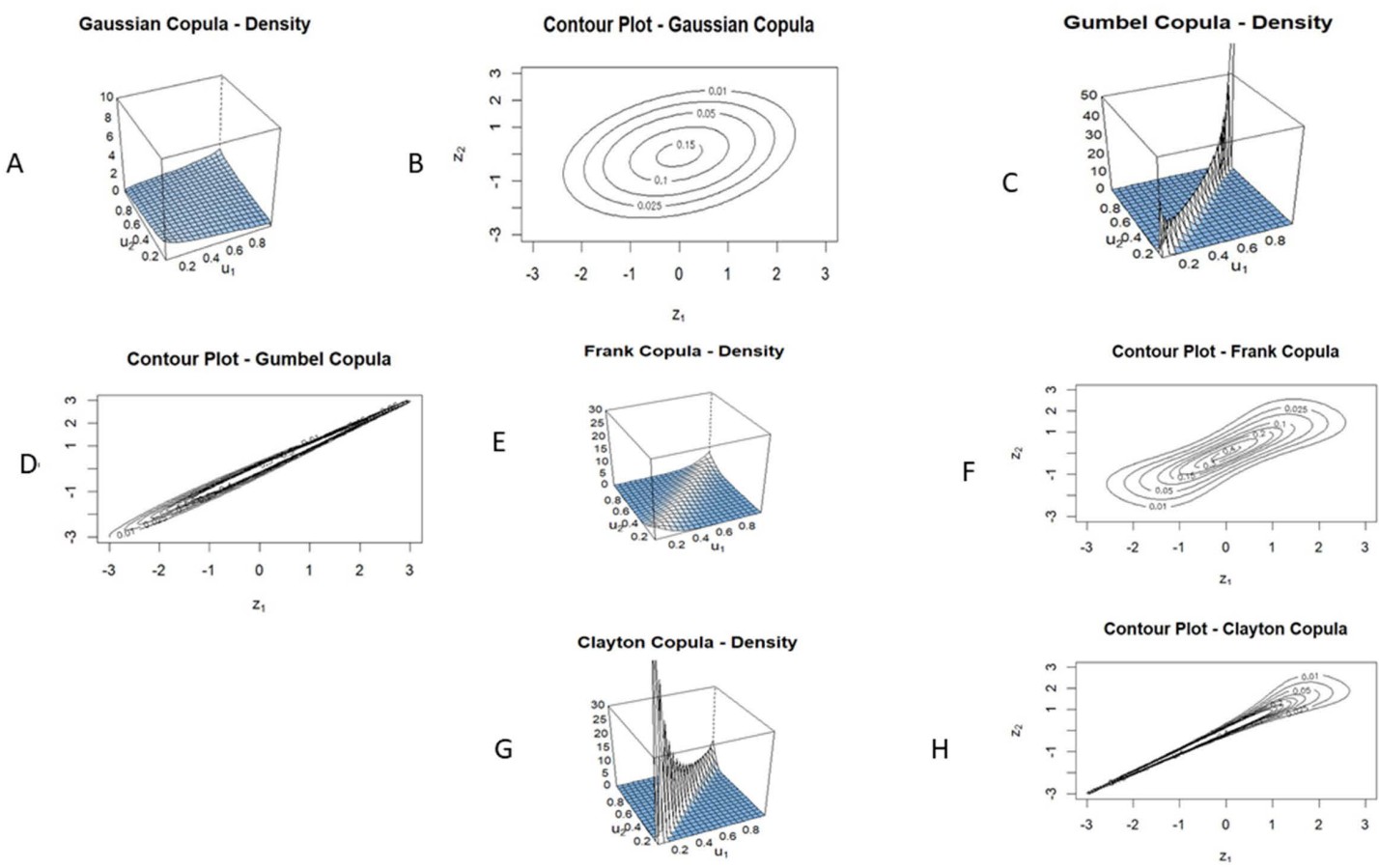

**Fig 8. Joint Density Surface and Contour Plots of Bivariate Copula Functions.** (A) Joint Density Surface of Gaussian Copula ($\theta = 0.99$, $\tau = 0.20$). (B) Contour Plot of Gaussian Copula. (C) Joint Density Surface of Gumbel Copula ($\theta = 17.00$, $\tau = 0.940$). (D) Contour Plot of Gumbel Copula. (E) Joint Density Surface of Frank Copula ($\theta = 35.00$, $\tau = 0.689$). (F) Contour Plot of Frank Copula. (G) Joint Density Surface of Clayton Copula ($\theta = 28.00$, $\tau = 0.933$). (H) Contour Plot of Clayton Copula. The copula parameter $\theta$ determines the shape of the joint density surface.

**Table 2. Comparison of Gaussian and Gumbel Copula Models.**

| Parameter | Gaussian Copula | | Gumbel Copula | |
|---|---|---|---|---|
| | **Estimate (SE)** | ***p*-value** | **Estimate (SE)** | ***p*-value** |
| Equation 1: **Overall Survival (Probit Link)** | | | | |
| Intercept | 4.649 (0.420) | <0.001*** | 4.702 (0.016) | <0.001*** |
| Age at Diagnosis | −0.085 (0.005) | <0.001*** | -0.080 (0.002) | <0.001*** |
| ER Status (Positive) | 0.765 (0.158) | <0.001*** | 0.387 (0.008) | <0.001*** |
| PR Status (Positive) | −0.196 (0.133) | 0.140 | −0.056 (0.021) | 0.007** |
| HER2 Status (Positive) | 0.005 (0.122) | 0.968 | −0.020 (0.072) | 0.775 |
| Chemotherapy (Yes) | 0.003 (0.168) | 0.986 | −0.043 (0.050) | 0.393 |
| Hormone Therapy (Yes) | −0.004 (0.121) | 0.974 | −0.007 (0.049) | 0.882 |
| Radiotherapy (Yes) | −0.005 (0.102) | 0.961 | −0.015 (0.037) | 0.687 |
| Nottingham Prognostic Index | −0.022 (0.064) | 0.734 | −0.037 (0.021) | 0.078. |
| Lymph Nodes Positive | −0.001 (0.016) | 0.952 | −0.000 (0.005) | 0.994 |
| Tumor Size (mm) | 0.003 (0.002) | 0.063. | 0.005 (0.002) | 0.008** |
| Equation 2: **Age at Diagnosis (Identity Link)** | | | | |
| Intercept | 60.605 (1.574) | <0.001*** | 60.774 (0.312) | <0.001*** |
| Overall Survival (Yes) | −7.898 (0.561) | <0.001*** | -5.312 (0.414) | <0.001*** |
| Histologic Grade 2 | 1.167 (1.088) | 0.283 | 0.295 (0.353) | 0.403 |
| Histologic Grade 3 | 0.385 (1.234) | 0.755 | 0.231 (0.147) | 0.114 |
| Nottingham Prognostic Index | −0.819 (0.371) | 0.027* | -0.627 (0.049) | <0.001*** |
| Tumor Size (mm) | 0.030 (0.019) | 0.118 | 0.049 (0.021) | 0.022* |
| Positive Lymph Nodes | −0.000 (0.085) | 0.998 | 0.003 (0.035) | 0.936 |
| ER Status (Positive) | 9.048 (0.789) | <0.001*** | 5.167 (0.428) | <0.001*** |
| PR Status (Positive) | −2.372 (0.650) | <0.001*** | -0.988 (0.376) | 0.009** |
| **Copula Parameters** | | | | |
| $\sigma^2$ (Variance) | 11.8 [11.4, 12.2] | | 12.6 [12.2, 13.1] | |
| $\theta$ (Dependence) | 0.999 [–] | | 17.0 [17.0, 17.0] | |
| Kendall's $\tau$ | 0.200 [0.180, 0.220] | | 0.940 [0.930, 0.950] | |
| AIC | 12,458 | | 11,927 | |
| BIC | 12,598 | | 12,067 | |

Significance codes: ***$p<0.001$, **$p<0.01$, *$p<0.05$, ·$p<0.1$.

The Gumbel copula indicates stronger dependence ($\tau = 0.940$) and better fit (lower AIC/BIC).

All models fitted on $n = 1,904$ patients from the METABRIC breast cancer cohort.

Tumor size is reported in mm for both descriptive and regression analyses.

$p < 0.001$), with survivors being nearly 8 years younger on average. This association reflects the joint dependence structure captured by the copula and should not be interpreted as survival explaining or causing age. The Gaussian copula parameter, Kendall's $\tau = 0.200$, suggests weak symmetric dependence between survival and age. The corresponding joint density and contour plots of Gaussian copula (Fig 8, Panel A and B), respectively, confirmed an elliptical, symmetric structure without strong tail dependency. The Gumbel model revealed a much stronger dependence structure, with a Kendall's $\tau = 0.940$, manifesting the dependence in the upper-tail. This indicates that exceptionally favorable outcomes, being alive at follow-up and having a younger age, tend to co-occur more frequently than expected under symmetric dependence. Statistically, the model improved fit over the Gaussian (AIC: 11,927 vs. 12,458). Certain covariates, such as ER and PR status, tumor size, and Nottingham Prognostic Index, were

statistically significant in both response variables. Particularly, tumor size emerged as a stronger predictor in the Gumbel model (Estimate = 0.005, $p = 0.008$).

Given the coding of the survival outcome (1 = alive), the positive tumor size coefficient indicates a higher modeled survival probability conditional on the joint dependence structure. This effect was not consistently observed across copula families (Tables 2–3), suggesting that it represents a model-specific conditional association rather than a robust marginal effect. The result is likely influenced by adjustment for correlated prognostic factors, such as treatment variables and the Nottingham Prognostic Index, together with the strong upper-tail dependence captured by the Gumbel copula. The joint density plot of the Gumbel copula (Fig 8, Panel C) and contour plot (Panel D) revealed a concentration of mass in

**Table 3. Comparison of Frank and Clayton Copula Models.**

| Parameter | Frank Copula | | Clayton Copula | |
|---|---|---|---|---|
| | Estimate (SE) | *p*-value | Estimate (SE) | *p*-value |
| Equation 1: **Overall Survival (Probit Link)** | | | | |
| Intercept | 4.722 (0.056) | <0.001*** | 4.655 (0.320) | <0.001*** |
| Age at Diagnosis | −0.082 (0.004) | <0.001*** | -0.082 (0.006) | <0.001*** |
| ER Status (Positive) | 0.700 (0.001) | <0.001*** | 0.558 (0.098) | <0.001*** |
| PR Status (Positive) | −0.170 (0.016) | <0.001*** | -0.095 (0.123) | 0.444 |
| HER2 Status (Positive) | −0.090 (0.006) | <0.001*** | 0.036 (0.088) | 0.680 |
| Chemotherapy (Yes) | −0.020 (0.027) | 0.453 | 0.036 (0.104) | 0.726 |
| Hormone Therapy (Yes) | 0.071 (0.042) | 0.086. | 0.007 (0.060) | 0.905 |
| Radiotherapy (Yes) | 0.030 (0.041) | 0.463 | 0.014 (0.055) | 0.799 |
| Nottingham Prognostic Index | −0.100 (0.036) | 0.006** | -0.042 (0.029) | 0.153 |
| Lymph Nodes Positive | −0.001 (0.024) | 0.958 | −0.005 (0.017) | 0.766 |
| Tumor Size (mm) | 0.003 (0.003) | 0.385 | 0.004 (0.013) | 0.790 |
| Equation 2: **Age at Diagnosis (Identity Link)** | | | | |
| Intercept | 61.370 (0.009) | <0.001*** | 60.970 (3.800) | <0.001*** |
| Overall Survival (Yes) | −8.530 (0.009) | <0.001*** | -6.300 (0.750) | <0.001*** |
| Histologic Grade 2 | 0.901 (0.001) | <0.001*** | 0.630 (0.395) | 0.111 |
| Histologic Grade 3 | 0.189 (0.001) | <0.001*** | 0.106 (0.288) | 0.714 |
| Nottingham Prognostic Index | −0.844 (0.000) | <0.001*** | -0.678 (0.197) | <0.001*** |
| Tumor Size (mm) | 0.026 (0.000) | <0.001*** | 0.042 (0.146) | 0.773 |
| Positive Lymph Nodes | −0.009 (0.000) | <0.001*** | 0.003 (0.176) | 0.987 |
| ER Status (Positive) | 8.812 (0.001) | <0.001*** | 6.403 (1.113) | <0.001*** |
| PR Status (Positive) | −2.236 (0.001) | <0.001*** | -0.967 (1.268) | 0.446 |
| **Copula Parameters** | | | | |
| $\sigma^2$ (Variance) | 11.9 [11.9, 11.9] | | 12.3 [12.0, 12.7] | |
| $\theta$ (Dependence) | 35.0 [35.0, 35.0] | | 28.0 [28.0, 28.0] | |
| Kendall's $\tau$ | 0.689 [0.680, 0.700] | | 0.933 [0.930, 0.940] | |
| AIC | 12,103 | | 11,939 | |
| BIC | 12,243 | | 12,079 | |

Significance codes: ***$p$<0.001, **$p$<0.01, *$p$<0.05, ·$p$<0.1.

The Clayton copula captures stronger lower-tail dependence and provides a better fit (lower AIC/BIC).

All models fitted on $n$ = 1,904 patients from the METABRIC breast cancer cohort.

Tumor size is reported in mm for both descriptive and regression analyses.

the upper-right quadrant, supporting the presence of upper-tail dependence. For further investigation of the dependence between overall survival and age at diagnosis, we employed Frank and Clayton copula models, both from the Archimedean family, and useful for modeling asymmetric and nonlinear dependencies.

In Table 3, the Frank copula model yielded a Kendall's $\tau$ = 0.689, suggesting moderate symmetric dependence between survival and age. Unlike the Gaussian copula, Frank is more flexible around the mode, enabling better modeling of the joint distribution when the dependence is strongest in the central range of the data.

In the survival model, age at diagnosis was negatively associated with survival (Estimate = −0.0819, $p < 0.001$), reaffirming the strong clinical relationship between younger age and better survival. ER-positive status had a high positive association (Estimate = 0.700, $p < 0.001$), while PR and HER2 statuses were significant but showed weak predictive value. In the second equation, survival status again showed a strong inverse relationship with age (Estimate = −8.53, $p < 0.001$), which should be interpreted as a symmetric association rather than reverse causality. Histologic grade 2 and ER/PR statuses remained significant. The density plot of the Frank copula (Fig 8, Panel E) revealed moderate clustering around the diagonal, consistent with central dependence. The contour plot (Panel F) showed symmetric contour lines that broaden from the centre, supporting the model's flexibility across the joint range. The Clayton copula exhibited a Kendall's $\tau$ = 0.933, indicating very strong lower-tail dependence. This suggests that unfavorable outcomes,older age at diagnosis and poor survival, tend to co-occur.

In the survival sub-model, age at diagnosis and ER status were again statistically significant. However, other covariates (PR status, HER2, and treatments) did not show significant associations. In the second equation, overall survival and ER status were significantly associated with age, whereas tumor size and lymph nodes did not contribute meaningfully. The density plot of the Clayton copula (Fig 8, Panel G) showed mass concentrated in the lower-left quadrant, highlighting dependence in the lower tail (older patients with poorer survival). The contour plot (Panel H) displayed asymmetric contour lines, with tighter spacing in the lower tail.

In summary, the Gumbel copula best captures favorable outcomes, not only in terms of goodness-of-fit and the lowest AIC and BIC, but also by exhibiting strong upper-tail dependence. The Clayton copula highlights joint risks in unfavorable outcomes, the Frank copula provides a balanced view of moderate symmetric dependence, and the Gaussian copula serves as a baseline model with limited ability to represent tail-dependent clinical patterns.

**Model diagnostics and comparison.** To evaluate whether the copula specification provides meaningful improvement over independent modeling, we compared the selected Gumbel copula model to separate marginal models fitted independently (probit for survival and Gaussian for age). The joint copula model demonstrated substantially better fit, with a likelihood ratio test strongly rejecting independence ($\chi^2$ = 2190.24, df = 1, $p < 0.0001$). The AIC decreased from 17,080 under the independent specification to 14,911 for the Gumbel copula ($\Delta$AIC = 2,168), and BIC decreased from 17,142–15,058 ($\Delta$BIC = 2,084). These large information-criterion differences provide overwhelming evidence that explicitly modeling dependence between survival and age improves overall model fit. Goodness-of-fit of the marginal distributions was further evaluated using Probability Integral Transform (PIT) diagnostics. Fig 9 presents the PIT results for both margins. For the continuous age margin, the PIT histogram displayed approximate uniformity (Kolmogorov–Smirnov test: $D$ = 0.027, $p$ = 0.126; Panel B), supporting the adequacy of the Gaussian marginal specification.

For the binary survival margin, a randomized PIT approach was used to account for discreteness [33]. The resulting randomized PIT values were approximately uniform (KS test: $D$ = 0.022, $p$ = 0.301; Panel A), indicating adequate specification of the probit marginal model. The corresponding QQ plots (Panels C and D) show close agreement with the uniform reference distribution.

Collectively, these diagnostics confirm that both marginal models are appropriately specified and that the copula framework captures additional dependence structure beyond what independent margins can represent.

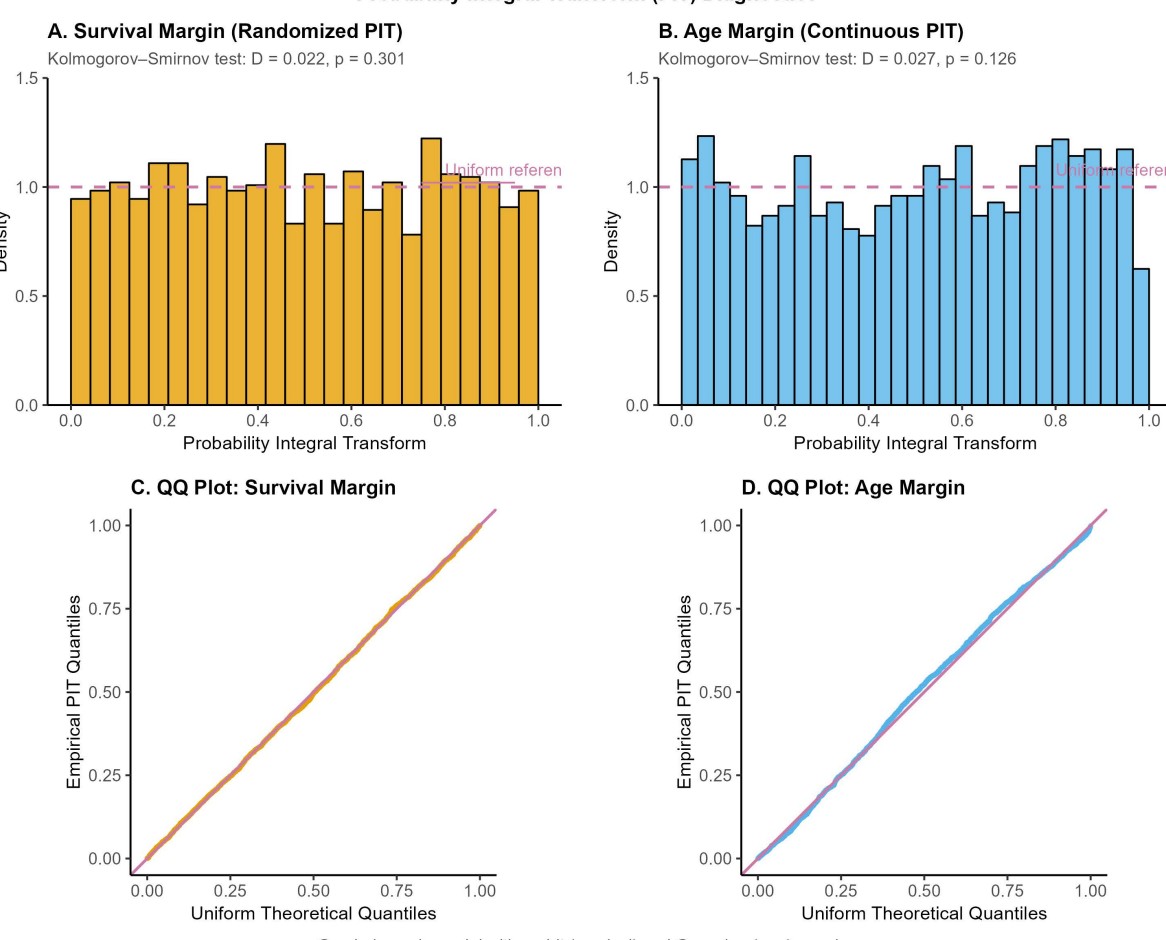

**Probability Integral Transform (PIT) Diagnostics**

Gumbel copula model with probit (survival) and Gaussian (age) margins

**Fig 9. Probability Integral Transform (PIT) diagnostics for the Gumbel copula model.** (A) Randomized PIT residuals for the binary survival margin. The histogram shows approximate uniformity (Kolmogorov–Smirnov test: $D = 0.022$, $p = 0.301$), indicating adequate probit specification. (B) PIT residuals for the continuous age margin (KS test: $D = 0.027$, $p = 0.126$), supporting the Gaussian marginal assumption. (C) QQ plot against the uniform distribution for the survival margin. (D) QQ plot against uniform distribution for age margin. The dashed red line represents uniform reference distribution in all panels.

## Conclusion

This study applied a copula-based modeling framework to jointly analyze survival status and age at diagnosis in breast cancer patients from the METABRIC cohort. The approach flexibly captured dependencies between the chosen pair of outcomes that traditional models fail to capture. Crucially, the copula formulation is symmetric and estimates statistical association rather than causal effects, addressing a common misinterpretation in joint models of temporally ordered outcomes. Among the copulas applied, the Gumbel copula showed the best fit to the data, effectively capturing the upper-tail dependence between younger age and better survival (as supported by AIC and BIC), while the Clayton copula highlighted the co-occurrence of poorer survival and older age through lower-tail dependence. Formal model comparison against an independent margins baseline confirmed that accounting for dependence via a copula significantly improves model fit (likelihood ratio test: $\chi^2 = 2190.24$, df = 1, $p < 0.0001$), and PIT diagnostics validated the adequacy of both marginal specifications.

The findings highlight that key clinical covariates, such as ER status, hormone therapy, and the Nottingham Prognostic Index, have a significant influence on the joint distribution of outcomes, whereas others (e.g., HER2 status or chemotherapy) showed limited or no association.

This study makes two key contributions: (i) it demonstrates the practical application of copula regression for mixed binary–continuous outcomes in a large, real-world cancer cohort, and (ii) it provides a reproducible template for model selection, diagnostic evaluation, and interpretation that can be adapted to other clinical settings.

Overall, this framework provides a robust and interpretable approach to exploring complex relationships in clinical data, and can be extended to include more outcomes or high-dimensional health studies. The analytical code and model specification details are provided in the Supporting Information to facilitate replication and extension by other researchers.

## Supporting information

**S1 Table. Comparison of AIC and BIC values for candidate marginal distributions of Age.** Normal, Gamma, Log-normal, Logistic, and Inverse-Gamma distributions were fitted to the data. The Normal distribution provided the best fit based on AIC and BIC.
(PDF)

**S1 Fig. Goodness of fit plots for fitted distributions to Age ($Y_2$).** Normal, Gamma, Log-normal, Logistic, and Inverse-Gamma distributions were fitted to the data. Visual inspection confirmed the Normal distribution as the most appropriate choice.
(TIFF)

**S2 Fig. Type of breast surgery and tumor laterality by survival status.** Combined bar plots depict the distribution of breast-conserving surgery versus mastectomy and tumor laterality (left vs. right breast) among surviving and deceased patients. Distinct patterns suggest potential associations between surgical choice, tumor location, and patient survival.
(TIFF)

**S3 Fig. Tumor cellularity across age groups.** Grouped bar charts show how cellularity (low, moderate, high) varies across age categories. Older patients more frequently exhibited higher cellularity levels, indicative of more aggressive disease profiles.
(TIFF)

**S4 Fig. Integrative molecular clusters across age groups.** Bar plots demonstrate how integrative molecular cluster frequencies vary across age categories. Differences in cluster prevalence with age suggest underlying biological and genomic heterogeneity within the cohort.
(TIFF)

**S5 Fig. Histologic subtype distribution across age groups.** The distribution of ductal/NST, lobular, and mixed histologic subtypes is shown by age category. While ductal/NST carcinoma remains predominant in all age groups.
(TIFF)

**S1 Appendix. Comprehensive Technical Details for the Copula Models.** This appendix provides the full technical derivation and computational details for the mixed binary-continuous copula models described in the main text.
(PDF)

## Author contributions

**Data curation:** Huma Rani.

**Formal analysis:** Huma Rani.

**Funding acquisition:** Laila A. Al-Essa.

**Methodology:** Huma Rani.

**Supervision:** Tahir Mehmood, Muhammad Aslam.

**Writing – original draft:** Huma Rani, Laila A. Al-Essa.

**Writing – review & editing:** Tahir Mehmood.

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
