## [Decision Letter · Decision Letter 0]

12 Jan 2026

Dear Dr. Rani,

Thank you for submitting your manuscript to PLOS ONE. After careful consideration, we feel that it has merit but does not fully meet PLOS ONE’s publication criteria as it currently stands. Therefore, we invite you to submit a revised version of the manuscript that addresses the points raised during the review process.

We look forward to receiving your revised manuscript.

Kind regards,

Yajie Zou

Academic Editor

PLOS One

Journal Requirements:

[Princess Nourah bint Abdulrahman University Researchers Supporting Project number

(PNURSP2025R443), Princess Nourah bint Abdulrahman University, Riyadh, Saudi

Arabia].

4. Thank you for uploading your study's underlying data set. Unfortunately, the repository you have noted in your Data Availability statement does not qualify as an acceptable data repository according to PLOS's standards.

At this time, please upload the minimal data set necessary to replicate your study's findings to a stable, public repository (such as figshare or Dryad) and provide us with the relevant URLs, DOIs, or accession numbers that may be used to access these data. For a list of recommended repositories and additional information on PLOS standards for data deposition, please see https://journals.plos.org/plosone/s/recommended-repositories....

5. Please ensure that you refer to Figure 5 in your text as, if accepted, production will need this reference to link the reader to the figure.

6. We notice that your supplementary figures are uploaded with the file type 'Figure'. Please amend the file type to 'Supporting Information'. Please ensure that each Supporting Information file has a legend listed in the manuscript after the references list.

Reviewers' comments:

Reviewer's Responses to Questions

**Comments to the Author**

1. Is the manuscript technically sound, and do the data support the conclusions?

Reviewer #1: Partly

Reviewer #2: Yes

2. Has the statistical analysis been performed appropriately and rigorously?

Reviewer #1: N/A

Reviewer #2: Yes

3. Have the authors made all data underlying the findings in their manuscript fully available?

Reviewer #1: No

Reviewer #2: Yes

4. Is the manuscript presented in an intelligible fashion and written in standard English?

Reviewer #1: Yes

Reviewer #2: Yes

Reviewer #1: Define the survival endpoint precisely and justify binarizing it. If survival time and censoring are available, consider a time to event formulation or clearly motivate why it’s not used.

Justify treating age at diagnosis as a response; otherwise, move age to the predictor side and focus the joint model on outcomes that are truly co modeled.

Remove or heavily qualify interpretations that look like survival “explains” age. Present Equation 2 as a symmetric association component (not causal) or restructure the model.

Correct the Kendall’s τ/θ inconsistency for the Clayton model and ensure the narrative matches Tables 2–3.

Clarify the full model specification, which covariates enter π, μ, σ², and θ. If θ is constant, remove claims about covariate dependent dependence (or report those results).

Reviewer #2: 1. How exactly is overall survival defined in the study and why reduce it to a binary outcome if time to event data exist?

2. Why is age at diagnosis treated as a response variable, given it temporally precedes survival?

3. If authors keep age as a response, how should readers interpret “Overall Survival (Yes)” predicting age in Equation 2 without implying reverse causality?

4. This study shows that Clayton Kendall’s τ = 0.60 for θ = 28, but Table 3 reports τ = 0.933.

5. This study states that the copula parameter θ is linked to covariates. Was θ actually modeled as covariate dependent, or is it constant (as the single θ per model suggests)?

6. Tumor size shows a positive association with “survival” in Table 2. Does this reflect coding, collinearity (e.g., with NPI), or scaling issues?

7. Beyond AIC/BIC, did the authors run any copula diagnostics (e.g., PIT/Rosenblatt type checks) or compare against a simple “independent margins” baseline to show what the copula adds?

8. Tumor size is reported in mm in Table 1 but appears as cm in the regression tables.

9. All figures should be replotted with better resolution.

10. The literature review lacks many applications of copula regression model for analysis and prediction in different fields. See:

Modeling car‐following behaviors using a driving style–based Bayesian model averaging Copula framework in mixed traffic flow.

Jointly analyzing freeway traffic incident clearance and response time using a copula-based approach.

.

Reviewer #1: No

Reviewer #2: No

---

## [Author Response · Author response to Decision Letter 1]

26 Feb 2026

We sincerely thank the Academic Editor and the Reviewers for their careful evaluation of our

manuscript and for their constructive and insightful comments. We have revised the manuscript

extensively to improve clarity, methodological transparency, and interpretative precision.

All changes have been incorporated in the revised manuscript and are highlighted in the marked up version. And all the changes have been detailed in the uploaded response to reviewer letter.

---

## [Decision Letter · Decision Letter 1]

19 Mar 2026

Unveiling the Multifaceted dynamics of breast cancer: A copula regression approach to modeling and predicting outcomes

PONE-D-25-56986R1

Dear Dr. Rani,

We’re pleased to inform you that your manuscript has been judged scientifically suitable for publication and will be formally accepted for publication once it meets all outstanding technical requirements.

Kind regards,

Yajie Zou

Academic Editor

PLOS One

Additional Editor Comments (optional):

Reviewers' comments:

Reviewer's Responses to Questions

**Comments to the Author**

Reviewer #1: (No Response)

Reviewer #2: (No Response)

2. Is the manuscript technically sound, and do the data support the conclusions?

Reviewer #1: (No Response)

Reviewer #2: (No Response)

3. Has the statistical analysis been performed appropriately and rigorously?

Reviewer #1: (No Response)

Reviewer #2: (No Response)

4. Have the authors made all data underlying the findings in their manuscript fully available?

Reviewer #1: (No Response)

Reviewer #2: (No Response)

5. Is the manuscript presented in an intelligible fashion and written in standard English?

Reviewer #1: (No Response)

Reviewer #2: (No Response)

Reviewer #1: This study examines the application of flexible copula regression models to analyze the

complex interdependencies among clinical variables in breast cancer data.

The authors have addressed all my concerns in the first of review.

Reviewer #2: My concerns have been addressed in the revised manuscript. The authors’ revisions have improved the clarity and completeness of the paper. I have no further comments for the authors.

.

Reviewer #1: No

Reviewer #2: No

---

## [Editor Report · Acceptance letter]

PONE-D-25-56986R1

PLOS One

Dear Dr. Rani,

I'm pleased to inform you that your manuscript has been deemed suitable for publication in PLOS One. Congratulations! Your manuscript is now being handed over to our production team.

Kind regards,

on behalf of

Dr. Yajie Zou

Academic Editor

PLOS One